# Keystone Species and Modularity in Microbial Hydrocarbon Degradation Uncovered by Network Analysis and Association Rule Mining

**DOI:** 10.3390/microorganisms8020190

**Published:** 2020-01-30

**Authors:** Florian Centler, Sarah Günnigmann, Ingo Fetzer, Annelie Wendeberg

**Affiliations:** 1Department of Environmental Microbiology, UFZ—Helmholtz Centre for Environmental Research, Permoserstraße 15, 04318 Leipzig, Germanyannelie.wendeberg@ufz.de (A.W.); 2Stockholm Resilience Centre, Stockholm University, Kräftriket 2B, 11419 Stockholm, Sweden

**Keywords:** hydrocarbon degradation, microbial communities, spatial scales, sample resolution, co-occurrence analysis, association rule mining, network analysis

## Abstract

Natural microbial communities in soils are highly diverse, allowing for rich networks of microbial interactions to unfold. Identifying key players in these networks is difficult as the distribution of microbial diversity at the local scale is typically non-uniform, and is the outcome of both abiotic environmental factors and microbial interactions. Here, using spatially resolved microbial presence-absence data along an aquifer transect contaminated with hydrocarbons, we combined co-occurrence analysis with association rule mining to identify potential keystone species along the hydrocarbon degradation process. Derived co-occurrence networks were found to be of a modular structure, with modules being associated with specific spatial locations and metabolic activity along the contamination plume. Association rules identify species that never occur without another, hence identifying potential one-sided cross-feeding relationships. We find that hub nodes in the rule network appearing in many rules as targets qualify as potential keystone species that catalyze critical transformation steps and are able to interact with varying partners. By contrasting analysis based on data derived from bulk samples and individual soil particles, we highlight the importance of spatial sample resolution. While individual inferred interactions are hypothetical in nature, requiring experimental verification, the observed global network patterns provide a unique first glimpse at the complex interaction networks at work in the microbial world.

## 1. Introduction

Natural soil teems with microbial life. Microbes are abundant and coexist as assemblages of extreme phylogenetic diversity [1]. This allows for rich networks of microbial interactions to unfold, ranging from antagonistic behavior, including prey–predator-type relationships [2], microbial warfare based on antibiotics production [3], and resource competition [4], to cooperative behavior, including the exchange of genetic material in horizontal gene transfer [5] and the exchange of metabolites required for growth [4]. The physical heterogeneity of the soil environment and the typically non-uniform distribution of organisms [6] add to the complexity of these systems. Microbial communities are self-organizing entities [7], which drive a wide range of processes of crucial importance for humanity, from global elemental cycles to the degradation of organic contaminants spilled locally, able to respond and adapt swiftly to environmental perturbations. Such responses can be surprisingly complex, for example, for petroleum hydrocarbon spillages reaching an aquifer, where transport and competition for terminal electron acceptors lead to the formation of spatial zones of alternating degradation pathways being active [8]. The full mineralization of aromatic hydrocarbons is usually achieved by teamwork, in which initial degraders do not mineralize hydrocarbons fully. Instead, they make partially mineralized compounds available to follow-up degraders, forming degradation cascades that can span several steps as in the reductive dechlorination of perchloroethylene (PCE) [9]. The phylogenetic diversity along such cascades needs to not be evenly distributed. While certain steps might be catalyzed by many taxa, other steps might only be catalyzed by few. In such a case, these few taxa qualify as keystone species whose presence is mandatory for the successful complete mineralization of hydrocarbon compounds. The comprehensive analysis of degradation activity in natural systems is hampered by the fact that only a small percentage of the natural diversity is cultivable [10], invalidating traditional methods of studying microbial interactions. Nevertheless, sequence-based fingerprinting methods targeting the 16S rRNA genes as a conserved phylogenetic marker [11] allow for the detection of microbes’ identity and their spatial distribution [12,13]. Applied to spatially resolved presence-absence data obtained at an aquifer contaminated with hydrocarbons, here, we introduce a combination of statistical and data mining methods as a novel approach to infer hypotheses on microbial interactions and dependencies [14]. The data analyzed here were not based on 16S rRNA gene amplicon sequencing but on terminal restriction fragment length polymorphism (T-RFLP) analysis, which does not provide the taxonomic resolution of the former but is a feasible approach to differentiate between archaeal and bacterial taxa. As it is important to sample at the spatial scale at which interactions occur [15], we contrasted samples derived from 1 g of soil with samples derived from individual soil particles.

## 2. Materials and Methods

### 2.1. Field Site and Data for Analysis

Data for analysis were obtained from the Leuna industrial site, Germany, which has been contaminated with megatons of petroleum hydrocarbons since World War II. Details on the site, sampling, and analytic procedures are given in [16]. Briefly, the capillary fringe and aquifer were sampled in five push cores of a 5-cm diameter and 1.2-m length. Each core was cut into sections of a 4-cm length, such that up to 16 depth levels ranging from 0 to 100 cm below the surface were covered. Four cores were located along the groundwater flow path (C1, C2, C3, and C5), spanning in total 510 m, with the contamination source located at core C1. A fifth downstream core C8 was hydrologically isolated from the plume from at least 1.5 years before sampling, and was sampled as a surrogate for pristine conditions. Two separate data sets were analyzed, differing in the spatial scale they cover. For the bulk data set, 1 g of soil was taken from 72 core sections of a 4-cm length distributed over all cores [16]. For the single particle data set, eight individual soil particles were extracted from each of 13 selected 4-cm core sections distributed over all cores, resulting in a total of 104 samples [17]. At least two core sections were chosen from each core, with one or two additional sections selected for cores 3 and 5, respectively. For both sample sets, consisting of 72 samples for the bulk set and 104 samples for the single particle set, DNA was extracted for each individual sample and purified. Bacterial and archaeal 16S rRNA gene fragments were subsequently amplified and digested either by restriction enzyme *Hae*III or *Rsa*I, followed by terminal restriction fragment length polymorphism (T-RFLP) analysis [16,17]. This procedure resulted in four presence-absence data sets, which were subsequently analyzed separately. Results for both restriction enzymes showed generally similar trends. Hence, only *Rsa*I results are discussed in the text and deviations briefly discussed, while *Hae*III results are presented in the Appendix A (Appendix A are analogous to Figure 1, Figure 2 and Figure 3).

### 2.2. Co-Occurrence Analysis

To test for correlations between the occurrence patterns of fragments, the exact Fisher test was applied to all fragment pairs. For this purpose, the distribution of two fragments was characterized by a 2-by-2 matrix, with the first row containing the number of samples in which both fragments were present and the number of samples containing only the first fragment, and the second row contains the number of samples only containing the second fragment and the number of samples without either fragment present. The Fisher test indicates the probability to obtain the observed, or a more extreme distribution pattern by chance. To correct for false positives due to multiple testing on the same data set, the Benjamini and Hochberg FDR controlling procedure [18] was subsequently applied to all *p*-values using the ’multtest’ library available for the statistical software package R (http://www.r-project.org). We identified significantly correlated fragment pairs as having a *p*-value below 0.05. To empirically estimate the number of false positives, we randomized the presence-absence matrix 100 times while maintaining row and column sums, hence preserving the observed fragment distributions, and applied the Fisher test to each randomization. To classify significantly correlated fragment pairs into positive correlations (co-occurring fragments) and negative correlations (fragments avoiding each other), we randomized the observed distribution patterns 1000 times maintaining the total number of occurrences for each fragment and computed the expected number of co-occurrences. Computing the probability that the observation is above or below this expectation with a *p*-value > 0.95 allowed us to classify significantly correlated pairs into cases of co-occurrence and mutual exclusion [19], using the ’vegan’ library in R.

### 2.3. Association Rule Mining

Presence-absence data were mined for association rules [20]. In particular, association rules of the form “fragment A→fragment B” were derived, which indicates that whenever fragment A was present in a sample, then fragment B was also present. We only considered rules with a confidence of 1, meaning that the data set did not contain a single exception to the rule. Rule mining was performed in R by iterating over all potential rules, requiring the inspection of all fragment pairs and two rules per fragment pair. For each potential rule, the data set was screened for the presence of an exception, in which case the rule was discarded. Only if no exception was found was the rule retained.

### 2.4. Network Analysis

Both co-occurrence relationships and association rules form networks with fragments as nodes and significant correlations or association rules defining undirected, respectively directed edges. Characteristics of the co-occurrence and rule networks, including the density, average number of neighbors, characteristic path lengths, clustering coefficients, centralization, and heterogeneity, were computed using NetworkAnalyzer [21] in Cytoscape [22], which was also used for the visualization and creation of intersections of the co-occurrence and rule networks. To test the co-occurrence networks for modularity, we computed the modularity index according to Newman [23] using the ’igraph’ package in R while only considering positive correlations. For a number of fragments, species could be assigned [16] and these are indicated in node labels. If an unambiguous assignment was not possible, all species names are given, separated by a slash.

## 3. Results

### 3.1. Co-Occurring Microbial Species Form Location and Function Specific Communities

Co-occurrence analysis identified between 250 and 26,984 significant correlations among fragments in all analyzed data sets (Table 1), with a false discovery rate of at most 0.07. Across all data sets, the majority of correlations were positive, indicating that the co-presence of species rather than their mutual exclusion was more common. While in bulk data sets, correlations in all possible combinations were present; between bacterial fragments, archaeal fragments, and between both types, in the single particle data sets, the vast majority (at least 94.8%) were correlations between bacterial fragments (Table 1). This difference was also evident when considering the hub nodes in the co-occurrence networks. These are nodes featuring a high network degree, taking part in many correlations: While hub nodes were typically archaeal fragments in the bulk data sets, bacterial fragments dominated the hub nodes in the particle data sets (Figure 1). Particle networks were larger, encompassing almost all detected fragments, and were better connected, featuring a higher density, a higher average number of neighbors, and shorter characteristic path lengths than the bulk data networks (Table 2). Bulk networks were of a modular structure with separated sub-networks, which were well-connected, referred to as modules from here on (Figure 2a). Closer analysis of these modules revealed a clear association between modules and locations along the contaminant plume (Figure 2b). An earlier study showed a clear functional separation of biogeochemical activity along the contaminant plume, with sulphate reduction taking place at the location of core C1, and methanogenesis at the locations of cores C2, C3, and C5, with only residual contaminant concentration levels being present at cores C3 and C5 [16]. Module 1 was dominated by bacterial fragments (Appendix A) and its presence restricted to sulphate-reducing conditions in core C1 and to a lesser extent to pristine conditions in core C8 (Figure 2b). It contained fragments identified as *Anaerolineaceae* and *Comamonadaceae*, which have both previously been associated with hydrocarbon degradation under sulphate-reducing conditions [24,25]. Module 2, also dominated by bacterial fragments, showed the broadest distribution, with the highest presence in cores C2, C3, and C5; a lesser presence in core C1; and a very low presence in the pristine core C8. In the case of *Hae*III digestion, modules 1 and 2 were not separated and the distribution pattern of *Hae*III’s module 1 (Appendix A) matched the combined pattern of *Rsa*I’s modules 1 and 2. Modules 3 and 4 were strongly dominated by archaeal fragments and only present in cores C3 and C5 (module 3) or solely in core C5 (module 4). *Acidovorax*, the only bacterial fragment in module 3, has been detected in methanogenic coal beds [26]. Finally, module 5 was mainly present in core C8, representing a pristine community. This clear spatial separation suggests that modules were part of communities that perform different functions. Initial sulphate reduction appears to be associated with module 1 and to a lesser extent with module 2. As mentioned, this separation was, however, not evident in the case of *Hae*III digestion. Module 2 was present in all methanogenic cores. Archaeal-dominated modules 3 and 4 were only present in one (module 4) or two methanogenic cores (module 3), where contaminant concentrations were already severely reduced.

### 3.2. Association Rule Networks Are Similar Across Spatial Sample Resolution but Differ in the Dominating Rule Type

An association rule of the form A→B indicates that whenever fragment A was present in a sample (the rule origin), then so was B (the rule target). Combining all detected association rules of this type results in an association rule network, whose properties were similar for bulk and single particle data sets (Table 3). Almost all fragments took part in at least one association rule. Networks were less dense as compared to the co-occurrence networks, and a clustering coefficient of zero indicated that two fragments taking part in association rules with the same fragment tended not to be directly linked in a separate association rule. Association rule networks were more hierarchic, with a greater tendency to be star-shaped and to contain hub nodes, as evidenced by higher values for centralization, heterogeneity, and shorter characteristic path lengths as compared to co-occurrence networks [27]. Between 1199 and 4560 association rules were derived from the data sets. In at most as 17.1% of these rules, the involved fragments were also found to be significantly correlated, indicating that association rules provide substantial additional information compared to co-occurrence analysis. When comparing the most common fragment type acting as the rule origin and target, the bulk and single particle data sets differed considerably (Table 4). While in bulk soil samples, both bacterial and archaeal fragments appeared as both the rule origin and rule target, the single particle rule network mainly consisted of rules of the type “archaeum → bacterium”, with at least 89.2%. This rule type was also common in the bulk data sets, albeit slightly trailing behind the rule type “bacterium → bacterium”.

### 3.3. Association Rule Network Hub Nodes Are of a Specific Type with a Characteristic Neighborhood

Association networks featured hub nodes with high node degrees, hence taking part in many association rules. When inspecting these fragments, a clear separation becomes apparent. Hub nodes either only appeared as rule targets or as rule origins. The former group, only appearing as rule targets, predominantly consisted of bacterial fragments, while archaeal fragments dominated hub nodes only appearing as rule origins. This separation was almost perfect for the single particle data sets and valid, with few exceptions for the bulk data sets (Figure 3 and Appendix A). To better characterize the interaction patterns of hub nodes appearing as rule targets, we analyzed the immediate association rule network neighborhood of hub node fragments and combined this with the co-occurrence network. Fragment types predicting the presence of the hub nodes differed between the bulk and the single particle data sets (Figure 4 and Appendix A). While both bacterial and archaeal fragments predicted hub nodes in the bulk data set, it was mainly archaeal fragments in the single particle data set. Some of the archaeal fragments were not co-occurring with other fragments in the network neighborhood, but others formed co-occurring sub-networks in both data sets.

## 4. Discussion

### 4.1. Limitations of Analyzed Data

Similar to OMICS data, T-RFLP data do not provide information on absolute quantities. Correlation analysis for relative data, however, requires advanced techniques [28,29]. By restricting our analysis to presence-absence data, we circumnavigated this problem at the cost of introducing a presence threshold. This can lead low abundant species to be mislabeled as absent in certain samples. Another limitation of T-RFLP data is the presence of unidentifiable fragments, fragments with ambiguous assignments, and multiple fragments potentially being assigned to the same species, which makes the biological interpretation of identified potential interactions difficult. While the problem of low taxonomic resolution can be solved by applying the introduced analysis technique to next-generation-sequencing data, such as 16S rRNA gene amplicon sequencing, for which the method is equally suited, the observed general network patterns are expected to be independent of the chosen data type.

### 4.2. Co-Occurrence Analysis and Association Rule Mining Address Different Types of Microbial Interactions

Co-occurrence indicates that two species have similar distribution patterns. This can be the result of a similar niche preference of both species. Competing for this niche, either competition is strong enough to only let one species survive at a location, or mild enough to allow for the coexistence of both species. Co-occurrence analysis detects both cases as mutual exclusion or co-occurrence. However, this case cannot be separated from the case in which both species directly interact, for example, by exchanging metabolic compounds. Regarding such direct interactions, co-occurrence analysis is only able to detect mutual interactions, leading to similar distribution patterns of both species as one species depends on the other and vice versa. A likely more prevalent interaction is the exchange of compounds in one direction in which the products of one species can be utilized by another. If this interaction is not exclusive, co-occurrence analysis cannot detect it. This is where association rule mining comes in handy, as it detects such one-sided dependencies. An association rule indicates that one species only occurs when another is present too. That is, the distribution of the first species is a subset of the distribution of the other. This might be a stronger indication for an actual interaction than the pure co-occurrence.

### 4.3. Interpreting Association Rules

Consider a case where each species contained in the set of producer species is able to degrade an initial substrate to a unique intermediary compound, which can further be metabolized by each species contained in a set of consumer species. If both sets include more than one species, and any species combination including at least one producer and one consumer species occurs in samples, then no association rule can be formulated. Only if there is an exclusive consumer or producer species will this species appear as a rule target and be associated with all species of the other set. Note that this same network motif of a hub species with incoming rules emerges in both cases. This indicates that the direction of a rule is not indicative of the direction of the dependency or the direction of the compound exchange: In the first case, the hub species is consuming compounds provided by several producer species, while in the second case, the hub species is providing a compound that is metabolized by several consumer species. To identify the directionality of a hypothesized metabolite exchange process, knowledge on the metabolic capacity of both species is helpful [30,31]. The presence of hub nodes with many incoming rules can be predicted by the presence of any of the species to which it is connected by a rule. Together with the exclusiveness, this makes hub nodes likely candidates for keystone species, which catalyze central steps along the degradation cascade. As fragments predicting the presence of a hub node were mostly not co-occurring, this indicates that the hub node interacted with alternative partners. Bacterial fragments are likely initial degraders while individual archaeal fragments are likely follow-up degraders in methanogenic and syntrophic relationships. Among the identifiable hub node fragments, *Sedimentibacter* and *Smithella* species were identified as keystone species candidates. This is consistent with *Smithella* already having been discussed as having a crucial role in hydrocarbon-contaminated ecosystems and playing a central role in syntrophic associations with methanogens [16,32,33].

### 4.4. Sampled Spatial Resolution Matters

For sampling the local scale, individual soil particles were subjected to analysis, using eight particles per location. It has been shown for marine sediments that four to eight single grains were sufficient to cover 50% of the OTU richness on the bulk level [34]. This indicates that our particle data were likely representative of the actual diversity present at the locations at the local scale. For the single particle data sets, co-occurrence relationships and association rules were dominated by a single type, namely bacterial fragments co-occuring with bacterial fragments, and rules pointing from archaeal fragments to bacterial fragments. On the local scale, bacterial competition hence did not seem to play a major role. This seems more to be the case for archaeal fragments. Although they did not feature many exclusion relationships (Table 1), the absence of co-occurrence relationships between them indicates that each sampled soil particle featured different archaeal fragments rather than a similar archaeal assemblage. In bulk soil samples, many single particles are mixed and analyzed together. Hence, even if single particles are populated by different archaea, in the bulk soil sample, they would appear to co-occur, explaining the many archaeal co-occurrence relationships detected for the bulk data set. Hub nodes only acting as rule targets were mainly bacterial fragments in all data sets, while hub nodes only acting as rule origins were dominated by archaeal fragments. This indicates that bacteria are more likely to be keystone species and able to interact with a wider variety of alternative interaction partners, as their presence can be predicted by many fragments. Archaeal fragments were more likely to predict the presence of a diverse sub-community containing many species, and hence likely be more dependent on the presence of other species. Potential interaction partners for keystone species candidates as indicated by appearing in an association rule with the keystone species were of both bacterial and archaeal origin in the bulk data set but predominantly of archaeal origin in the particle data set. This might be caused by the typical spatial distance of the interaction, which can be longer for compounds, such as acetate, to be passed from an initial degrader to a follow-up degrader, while, for example, syntrophic H_2_ exchange with an archaeal methanogen requires close spatial proximity. While the bulk data sets capture both interaction types, the particle data set only captures the latter interaction type.

### 4.5. Inferred Degradation Cascade Architecture

It has been suggested that habitat filtering is the major factor determining community composition in the natural environment and the human microbiome [35,36]. We found this not to be true for our hydrocarbon-contaminated aquifer. For less than half of all fragments, the environmental data was good enough to predict the presence of the fragment (Appendix A), indicating that factors beyond habitat filtering, such as species interactions, played an essential role. This might have been triggered by the hydrocarbon input, which provokes a degradation cascade requiring multiple microbial interactions. Although statistical and data mining methods only indicate potential interactions between microbial species, the global patterns formed by them and evidenced in the resulting network structures provide profound insights regarding community assembly and the interaction of metabolically specialized species [37]. Co-occurrence networks derived from the bulk data set uncovered a modular structure of microbial degradation activity of distinct, spatially separated sub-communities. Interestingly, this structure was not visible when analyzing the single particle data, indicating a higher variability of community composition at the local scale, where heterogeneities might be a more influential factor. While topological parameters of co-occurrence networks have been used to identify keystone species [38], here, we used association rule networks, which have the benefit of being able to uncover one-sided dependencies. Association rule networks were more robust than co-occurrence networks with respect to the spatial scale. Topological indices were more similar for them when comparing bulk and single particle data sets. Likewise, rule hub nodes had similar characteristics. Keystone species candidates as hub nodes only acting as rule targets were predominantly of bacterial origin. Ubiquitous species, present in all samples, would appear as trivial and ultimate hub nodes, with incoming rules from all detected fragments. This was not the case in our data, in which most connected hub nodes had at most incoming links from half of all detected fragments. However, the higher occupancy of bacteria than archaea contributes to the dominance of bacteria as rule targets (Appendix A). If keystone species catalyze a critical step that is sandwiched between steps that are performed by many taxa, a bow tie structure emerges: Many initial degraders provide compounds for the keystone species, which provides compounds for a variety of downstream degraders. Such a bow tie concept has been proposed for intracellular metabolic transformation networks and might be a general principle of biological systems [39,40]. Within cells, a broad variety of substrate compounds are converted to a handful of key metabolites, which are then assembled to create the full diversity of compounds necessary for cellular growth and maintenance.

## 5. Conclusions

Co-occurrence analysis and data mining were used in this study to uncover the organization of the degradation activity of natural microbial communities in response to hydrocarbon input in an aquifer. Both approaches address different types of interactions. While correlation analysis is the more commonly applied method, we highlighted its limitations and demonstrated the benefit of association rule mining, which is able to identify one-sided dependencies, non-exclusive interactions, as well as likely keystone species candidates. We furthermore showed that the spatial resolution of samples is important and should be adjusted to the typical length of the interaction one wishes to investigate. To corroborate or dismiss identified potential interactions, additional information on the metabolic potential of involved species is necessary and should be included in the analysis, for example, that derived from accompanying metagenomic, metatranscriptomic, or metaproteomic data. Cultivation-independent methods are currently amassing tremendous data sets of microbial communities inhabiting natural environments and the microbiomes of higher organisms. The use of statistical and data mining methods is an easily applied first step to untangle the tremendous network of diverse microbial interactions at play in microbial communities.

## Figures and Tables

**Figure 1 microorganisms-08-00190-f001:**
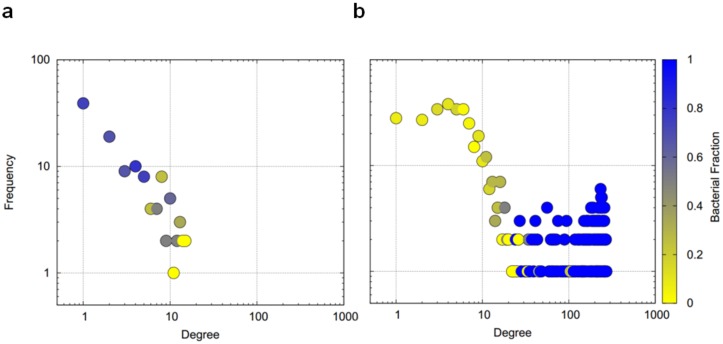
Degree distribution of co-occurrence networks. Color indicates the fraction of bacterial fragments at the respective degree levels, ranging from yellow indicating archaeal fragments only to blue indicating bacterial fragments only. Networks are based on *Rsa*I digestion. (**a**) Network derived from bulk soil samples; (**b**) Network derived from single particle samples.

**Figure 2 microorganisms-08-00190-f002:**
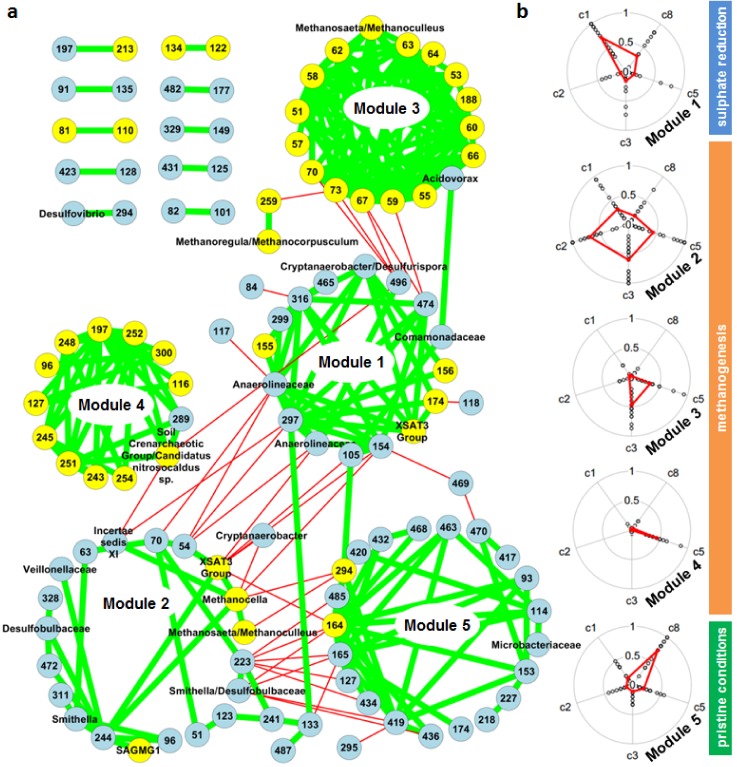
The co-occurrence network contains five modules whose presence varies along the flow path. The network is based on bulk soil samples and *Rsa*I digestion. (**a**) Bacterial fragments (light blue nodes) and archaeal fragments (yellow nodes) are connected by green edges for positive (co-occurrence) and thin red edges for negative correlations (mutual exclusion). Fragments belonging to modules as discovered by network analysis are arranged on circles; (**b**) Modules’ presence along the flow path (cores C1, C2, C3, and C5) and in the pristine environment (C8) is shown in radial plots detailing the presence of individual module fragments (symbols) and module mean (red line), with values ranging from 0 (fragment not present in any core sample) to 1 (fragment present in all core samples). According to geochemical data, sulphate reduction was limited to core C1, while methanogenesis took place in cores C2, C3, and C5.

**Figure 3 microorganisms-08-00190-f003:**
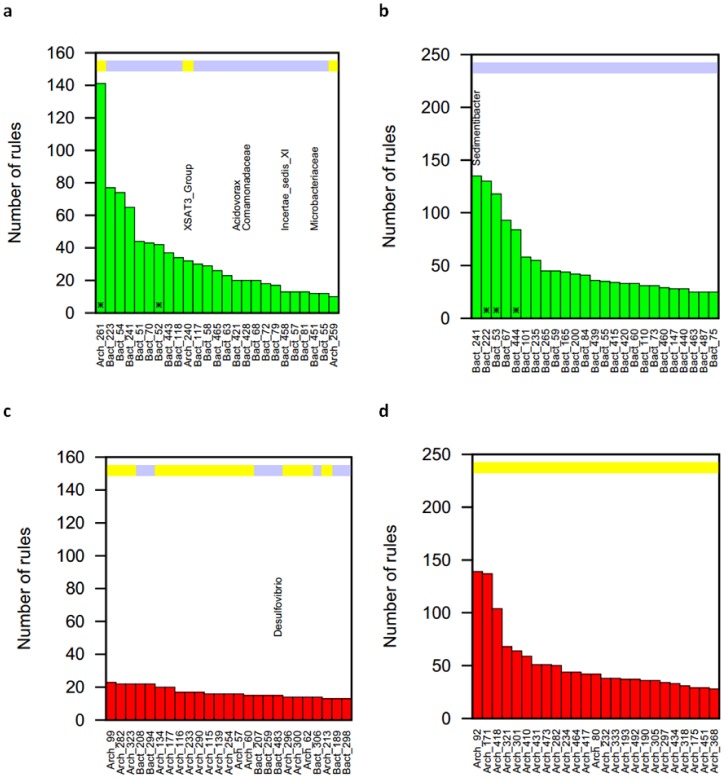
Hub nodes in association rule networks. Top 25 fragments exclusively participating in rules as targets or origins in the bulk and single particle data sets. Color bands indicate the fragment type as bacterial (light blue) or archaeal (yellow). Networks are based on *Rsa*I digestion. (**a**) Hub fragments acting as rule targets in the bulk data set; (**b**) Hub fragments acting as rule targets in the single particle data set; (**c**) Hub fragments acting as rule origins in the bulk data set; (**d**) Hub fragments acting as rule origins in the single particle data set. Unambiguously identified fragments are noted, and bars marked with “*” had ambiguous identities: Arch_261: Methanoregula/SAGMG1/Marine Group 1 Archaea; Bact_52: Smithella/Desulfobulbaceae; Bact_222: Smithella/Dehalobacter/Desulfobulbaceae/Clostridiales; Bact_53: Smithella/Desulfobulbaceae; Bact_444: Proteiniphilum/Anaerolineaceae.

**Figure 4 microorganisms-08-00190-f004:**
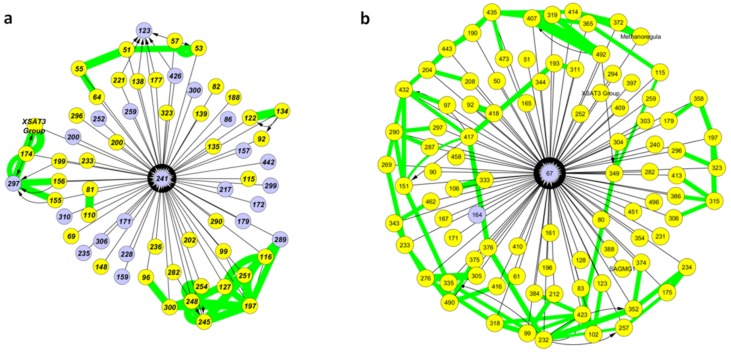
Selected representative hub nodes of the association rule networks only participating as rule targets. Showing all rule associations involving the central hub node fragment and directly neighboring fragments (arrows), and overlaying the co-occurrence network with green edges indicating co-occurrence relationships. Bacterial fragments are shown in light blue and archaeal fragments in yellow. Networks are based on *Rsa*I digestion. (**a**) The top fourth hub node of the bulk data set; (**b**) The top fourth hub node of the single particle data set (B).

**Table 1 microorganisms-08-00190-t001:** Significantly correlated fragments.

	Bulk		Single Particle	
Samples	72		104	
Restriction Enzyme	*Rsa*I	*Hae*III	*Rsa*I	*Hae*III
**Fragments Total**	274	351	627	612
(Bacteria/Archaea)	(201/73)	(230/121)	(329/298)	(326/286)
**Potential Pairs**	37,401	61,425	196,251	186,966
**Correlated Pairs** ^1^	250 (0.7%)	1906 (3.1%)	26,984 (13.7%)	24,150 (12.9%)
**On Randomized Data** ^2^	0.02 (0.0%)	0.07 (0.0%)	1811.2 (0.9%)	1674.3 (0.9%)
**FDR (estimated)**	< 1 × 10^−4^	< 1 × 10^−4^	0.07	0.07
**Positive Correlations** ^3^	212 (84.8%)	1781 (93.4%)	25,739 (95.4%)	23,476 (97.2%)
**Negative Correlations** ^3^	38 (15.2%)	125 (6.6%)	311 (1.2%)	265 (1.1%)
**Bacterial Pairs**	91 (36.4%)	427 (22.4%)	25,577 (94.8%)	22,910 (94.9%)
(Pos./Neg. Correlations)	(73/18)	(372/55)	(24,577/69)	(22,450/56)
**Archaeal Pairs**	106 (42.4%)	1326 (69.6%)	723 (2.7%)	460 (1.9%)
(Pos./Neg. Correlations)	(102/4)	(1323/3)	(695/28)	(447/13)
**Mixed Pairs**	53 (21.2%)	153 (8.0%)	684 (2.5%)	780 (3.2%)
(Pos./Neg. Correlations)	(37/16)	(86/67)	(467/214)	(579/196)

^1^ Significant correlations between two fragments according to the exact Fisher test (*p* < 0.05) with correction for multiple testing; ^2^ mean over 100 permutations maintaining row and column sums; ^3^ Pairs occurring more often (positive) or less often (negative) together compared to random expectation given by the observed distributions of both fragments (with *p* > 0.95).

**Table 2 microorganisms-08-00190-t002:** Topological parameters of co-occurrence networks.

	Bulk		Single Particle	
Restriction Enzyme	*Rsa*I	*Hae*III	*Rsa*I	*Hae*III
**Network Coverage** ^1^	43%	75%	98%	95%
**Number of Nodes/Edges**	118/250	264/1906	615/26,984	579/24,150
**Density**	0.036	0.055	0.14	0.14
**Avg. Number of Neighbors**	4.24	14.44	87.75	83.42
**Characteristic Path Length**	4.27	3.29	2.65	2.85
**on random network** ^2^	3.41	2.37	1.86	1.86
**Clustering Coefficient**	0.36	0.37	0.48	0.47
**on random network** ^2^	0.03	0.06	0.14	0.14
**Centralization**	0.094	0.19	0.30	0.30
**Heterogeneity**	0.90	1.15	1.11	1.13
**Modularity** ^3^	0.77	0.39	0.044	0.044
**Number of Modules** ^3^	17	9	(6)	(11)
**with more than 3 fragments**	5	4	(4)	(7)

^1^ Percentage of total fragments present in co-occurrence network; ^2^ average over 100 Erdös–Rényi random networks with prescribed number of nodes and edges; ^3^ Computed only considering positive interactions.

**Table 3 microorganisms-08-00190-t003:** Topological parameters of the association rule networks.

	Bulk		Single Particle	
Restriction Enzyme	*Rsa*I	*Hae*III	*Rsa*I	*Hae*III
**Network Coverage** ^1^	92%	95%	92%	100%
**Number of Nodes/Edges**	253/1199	335/2214	576/2834	612/4560
**Density** ^2^	0.032	0.027	0.017	0.022
**Avg. Number of Neighbors**	8.12	9.06	9.52	13.66
**Characteristic Path Length**	1.15	1.35	1.03	1.08
**Clustering Coefficient**	0.0	0.0	0.0	0.0
**Centralization** ^2^	0.34	0.28	0.22	0.49
**Heterogeneity** ^2^	1.22	1.18	1.68	1.75

^1^ Percentage of total fragments present in rule network; ^2^ Computed ignoring edge directionality.

**Table 4 microorganisms-08-00190-t004:** Association rule networks.

	Bulk		Single Particle	
Restriction Enzyme	*Rsa*I	*Hae*III	*Rsa*I	*Hae*III
**Association rules**	1199	2214	2834	4560
**also sign. correlated**	49 (4.1%)	379 (17.1%)	112 (4.0%)	186 (4.1%)
**Bacterium→Bacterium**	468 (39%)	768 (34.7%)	11 (0.4%)	371 (8.1%)
**Bacterium→Archaeum**	126 (10.5%)	107 (4.8%)	0 (0%)	0 (0%)
**Archaeum→Archaeum**	145 (12.1%)	698 (31.5%)	49 (1.7%)	123 (2.7%)
**Archaeum→Bacterium**	460 (38.4%)	641 (29%)	2774 (97.9%)	4066 (89.2%)

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
