# Peer review of "Keystone Species and Modularity in Microbial Hydrocarbon Degradation Uncovered by Network Analysis and Association Rule Mining"

_microorganisms, 2020, doi:10.3390/microorganisms8020190_

Round 1
Reviewer 1 Report
The manuscript „Keystone species and modularity in microbial hydrocarbon degradation uncovered by network analysis and association rule mining“ by Centler et al. presents the results of co-occurrence analysis and data mining of the microbial communities organization within a soil contaminated with hydrocarbons. Applied association rule mining allowed authors to identify probable keystone species candidates inhabiting the examined environment. By extension, application of such analysis to data sets of various microbial communities might become crucial for understanding the complex interaction networks of microbiome.
Overall the manuscript is well written. The only limitation is a quite low resolution of supplemental data.
Author Response
"Overall the manuscript is well written. The only limitation is a quite low resolution of supplemental data."
Thank you very much for pointing out the low resolution in the supplemental data. We identified a problem during PDF conversion which has now been fixed.
Reviewer 2 Report
In their manuscript, Centler et al analyze microbial association/co-occurence in a hydrocarbon-contaminated site, at various scales, to identify potential interspecies and environmental interactions. I like the fundamental concept but unfortunately using TRFLP as a technique to answer those questions is quite outdated and unable not only to provide relative abundance data but lack the necessary taxonomic resolution, especially for uncultured taxa. On top of that the data cannot be compared across studies, so its unclear how the results will drive subsequent studies. Amplicon sequencing has become a widely used and cost effective approach and is quite accessible, so the decision to use TRFLP is hard to justify.
The authors should also have included some minimal information about the samples, like how many samples, how many categories etc, rather than send the reader to
a paper published 6 years ago.
Author Response
"In their manuscript, Centler et al analyze microbial association/co-occurence in a hydrocarbon-contaminated site, at various scales, to identify potential interspecies and environmental interactions. I like the fundamental concept but unfortunately using TRFLP as a technique to answer those questions is quite outdated and unable not only to provide relative abundance data but lack the necessary taxonomic resolution, especially for uncultured taxa. On top of that the data cannot be compared across studies, so its unclear how the results will drive subsequent studies. Amplicon sequencing has become a widely used and cost effective approach and is quite accessible, so the decision to use TRFLP is hard to justify."
We fully agree with the severe limitations of TRFLP data as compared to16S rRNA gene based amplicon sequencing. We note however, that this approach was sufficient to discriminate between archaeal and bacterial species which led to interpretable results in context of the derived network structures. We believe that the obtained and discussed general network properties of both co-occurrence and rule networks are independent of the used technique for detecting the presence (or absence) of individual taxa. We agree however, that the full potential of our new approach can only by realized when applied to spatially resolved data of high taxonomic resolution which was, unfortunately, not available for the test site. As obtained taxonomic results are indeed difficult to compare across study, we rather focus on the introduction of our new methodology and promote its application to future data sets.
We added clarifying statements in the revised manuscript (P. 2, L. 55-62;P. 9, L. 257-261) in track-changes-enabled manuscript).
"The authors should also have included some minimal information about the samples, like how many samples, how many categories etc, rather than send the reader to a paper published 6 years ago."
We added some more details regarding the samples (see changes in Section 2.1, P. 2, L. 66-87).
Round 2
Reviewer 2 Report
I dont have further suggestions or requests